# Implantable cardioverter defibrillator therapy is cost effective for primary prevention patients in Taiwan: An analysis from the Improve SCA trial

**Reece Holbrook**[1], **Lucas Higuera**[1], **Kael Wherry**[1], **Dave Phay**[1], **Yu-Cheng Hsieh**[2], **Kuo-Hung Lin**[3], **Yen-Bin Liu**[4]*

1 Medtronic, plc, Dublin, Ireland, 2 Department of Cardiology, Taichung Veterans General Hospital and National Yan-Ming University School of Medicine, Taichung, Taiwan, 3 Department of Cardiology, China Medical Center University Hospital, 4 Division of Cardiology, Department of Internal Medicine, National Taiwan University Hospital, Taipei, Taiwan

* yenbin@ntu.edu.tw

**Data Availability Statement:** All relevant data are within the manuscript and its Supporting Information files.

## Abstract

### Objective

Implantable cardiac defibrillators (ICDs) for primary prevention (PP) of sudden cardiac arrest (SCA) are well-established but underutilized globally. The Improve SCA study has identified a cohort of patients called 1.5 primary prevention (1.5PP) based on PP patients with the presence of certain risk factors. We evaluated the cost-effectiveness of ICD therapy compared to no ICD among the PP population and the subset of 1.5PP patients in Taiwan.

### Methods

A Markov model was run over a lifetime time horizon from the Taiwan payer perspective. Mortality and utility estimates were obtained from the literature (PP) and the IMPROVE SCA trial (1.5PP). Cost inputs were obtained from the Taiwan National Health Insurance Administration (NHIA), Ministry of Health and Welfare. We used a willingness-to-pay (WTP) threshold of NT$2,100,000, as established through standard WTP research methods and in alignment with World Health Organization recommendations.

### Results

The total discounted costs for ICD therapy and no ICD therapy were NT$1,664,259 and NT$646,396 respectively for PP, while they were NT$2,410,603 and NT$905,881 respectively for 1.5PP. Total discounted QALYs for ICD therapy and no ICD therapy were 6.48 and 4.98 respectively for PP, while they were 10.78 and 7.71 respectively for 1.5PP. The incremental cost effectiveness ratio was NT$708,711 for PP and NT$441,153 for 1.5PP, therefore ICD therapy should be considered cost effective for PP and highly cost effective for 1.5PP.

**Funding:** Medtronic provided support for this study in the form of salaries for RH, LH, KW, and DP. The specific roles of these authors are articulated in the 'author contributions' section. The funders had no role in study design, data collection and analysis, decision to publish, or preparation of the manuscript.

**Competing interests:** The authors have read the journal's policy and the authors have the following competing interests: YBL and YCH received speaker fees/steering committee fees from Medtronic; YCH and KHL were co-investigators of the Medtronic Improve SCA trial; RH, LH, KW, and DP are paid employees of Medtronic. These employees of Medtronic are also shareholders. This does not alter our adherence to PLOS ONE policies on sharing data and materials. There are no patents, products in development or marketed products associated with this research to declare.

## Conclusions

ICD therapy compared to no ICD therapy is cost-effective in the whole PP population and highly cost-effective in the subset 1.5PP population in Taiwan.

## Introduction

Evidence for the use of implantable cardioverter defibrillators (ICDs) for primary prevention of sudden cardiac arrest (SCA) in patients with moderately symptomatic heart failure and reduced systolic function is well-established through multiple randomized clinical trials [1, 2] and confirmed in real-world observational evidence [3, 4]. This evidence has led to strong recommendations for ICD use in society guidelines [5, 6] and has been leveraged to establish the cost-effectiveness of ICD therapy in multiple healthcare systems [7, 8]. Despite this strong evidence base, ICD therapy remains underutilized globally, due at least in part to cost considerations and lack of reimbursement [9].

The Improve SCA study has identified a high-risk subset of primary prevention patients called 1.5 primary prevention based on the presence of at least one of the following documented risk factors: non-sustained ventricular tachycardia (NSVT), frequent premature ventricular contractions (PVCs) >10/h, left ventricular ejection fraction (LVEF) <25%, presyncope or syncope [10]. Improve SCA patients with 1.5 primary prevention characteristics were found to have a higher rate of treatment with appropriate therapy than primary prevention patients, and when treated with an ICD, 1.5 primary prevention patients experienced a 49% relative risk reduction in all-cause mortality.

While the cost-effectiveness of ICD therapy for primary prevention patients has been established in western countries, it has not been previously established for the healthcare system in Taiwan. Furthermore, the cost-effectiveness of ICD therapy for 1.5 primary prevention patients is not well known. The 1.5 primary prevention cohort could be used to prioritize health care resources in geographies where such resources are insufficient to cover the full primary prevention population. The aim of the present study was to critically evaluate the cost-effectiveness models of ICD therapy for both the superset of primary prevention and the subset 1.5 of primary prevention patients with heart failure in the Taiwan healthcare system, and to identify the main factors influencing the cost-effectiveness of ICD therapy [11].

## Methods

We used an existing Markov decision model [7] to estimate the lifetime cost, quality of life, survival, and incremental cost-effectiveness of ICD therapy versus no ICD therapy for a Taiwanese population at risk for SCA (both primary prevention and 1.5 primary prevention). The Improve SCA study [11] protocol was approved by the Institutional Review Board or Medical Ethics Committee of each respective study center. This analysis is a modeling exercise based on previously published data from the Improve SCA study [11] and does not involve any additional human research. No ICD therapy was selected as the control instead of pharmacologic therapy based SCD-HeFT study findings that indicated no significant difference in the risk of death between treatment with amiodarone and treatment with a placebo [1]. Model inputs are shown in Table 1 and described in detail, below. The model was implemented in Microsoft Excel, as described previously [7].

**Table 1. Model input parameters.**

| Model Parameters | Base Case Value | Standard Error | Distribution | Reference |
|---|---|---|---|---|
| **Monthly Risk of Mortality (ICD Therapy 1.5PP)** | | | | |
| Sudden cardiac death | 0.0007 | 0.0003 | Beta | [11] |
| Non-sudden cardiac death | 0.0014 | 0.0004 | Beta | |
| Non-cardiac death | 0.0005 | 0.0003 | Beta | |
| Unknown death | 0.0013 | 0.0003 | Beta | |
| **Monthly Risk of Mortality (No ICD Therapy 1.5PP)** | | | | |
| Sudden cardiac death | 0.0028 | 0.0005 | Beta | [11] |
| Non-sudden cardiac death | 0.0021 | 0.0004 | Beta | |
| Non-cardiac death | 0.0010 | 0.0004 | Beta | |
| Unknown death | 0.0014 | 0.0004 | Beta | |
| **Monthly Risk of Mortality, ICD Therapy (Primary Prevention)** | | | | |
| Sudden cardiac death | 0.0015 | 0.0001 | Beta | |
| Heart failure death | 0.0029 | 0.0002 | Beta | |
| Other cardiac death | 0.0004 | 0.00002 | Beta | |
| Non-cardiac death | 0.0024 | 0.0002 | Beta | |
| **Monthly Risk of Mortality, No ICD Therapy (Primary Prevention)** | | | | |
| Sudden cardiac death | 0.0042 | 0.0004 | Beta | [7] |
| Heart failure death | 0.0029 | 0.0003 | Beta | |
| Other cardiac death | 0.0002 | 0.00002 | Beta | |
| Non-cardiac death | 0.0031 | 0.0003 | Beta | |
| **ICD-Related Probabilities** | | | | |
| Initial operative death | 0.0002 | 0.00002 | Beta | [12] |
| Continue ICD therapy after shock | 0.0034 | 0.0002 | Beta | [13–17] |
| Discontinue ICD therapy after shock | 0.0001 | 0.00007 | Beta | |
| Lead replacement (initial implant) | 0.0004 | 0.0005 | Beta | [18, 19] |
| Lead replacement (replacement implant) | 0.0008 | 0.0009 | Beta | [20] |
| Lead dislodgement (initial implant) | 0.018 | 0.0012 | Beta | |
| Lead dislodgement (replacement implant) | 0.005 | 0.0009 | Beta | [13] |
| ICD infection (initial implant) | 0.0244 | 0.0049 | Beta | [21] |
| ICD infection (replacement implant) | 0.0432 | 0.0064 | Beta | [22] |
| **Costs, 2017 New Taiwan Dollar (NT$)** | | | | |
| ICD implant procedure (initial) | NT$633,678 | | | [23] |
| ICD implant procedure (replacement) | NT$315,513 | | | |
| Lead replacement | NT$62,757 | | | |
| ICD infection | NT$765,213 | | | |
| ICD lead dislodgement | NT$61,566 | | | |
| ICD generator removal | NT$64,290 | | | |
| ICD inappropriate shocks | NT$670 | | | [24] |
| Monthly inpatient cost | NT$6,828 | | | [25] |
| Monthly outpatient cost | NT$858 | | | |
| Discount rate | 1.375% | | | [26] |
| **Utility Primary Prevention** | | | | |
| Annual utility | 0.7315 | 0.0126 | Beta | [14] |
| ICD complication state | 0.6474 | 0.112 | Beta | |
| **Utility 1.5 Prevention** | | | | |

(*Continued*)

**Table 1.** (Continued)

| Model Parameters | Base Case Value | Standard Error | Distribution | Reference |
|---|---|---|---|---|
| Annual utility | 0.8683 | 0.0360 | Beta | [27] |
| ICD complication state | 0.7685 | 0.0360 | Beta | |

**Abbreviations:** ICD, Implantable Cardioverter-Defibrillator

## Model structure

The model is structured as a decision tree with two treatment arms, ICD therapy or no ICD therapy, followed by consecutive Markov models (Fig 1). The model was run in two separate scenarios, following a simulated cohort of 1,000 patients with a standard indication for primary prevention ICD therapy, and for patients with a standard indication for primary prevention ICD therapy with one or more 1.5 primary prevention risk factors. Patients who enter the model in the ICD arm are at an initial risk of operative death or survival. Patients who survive the ICD surgery enter the Markov model in the well state. From the well state, ICD patients stay well or progress to ICD complications, sudden cardiac death, non-sudden cardiac death, non-cardiac death, unknown death. Patients remain in the same state or progress to a different state at the beginning of each cycle, except for the complication state. Patients who experience an ICD complication remain in the complication state for only one cycle, then progress to continued ICD therapy or discontinued ICD therapy. In the event of therapy discontinuation, ICD patients stay well without ICD treatment or progress to sudden cardiac death, non-sudden cardiac death, non-cardiac death, unknown death. Patients in the no ICD arm enter the model in the well state and remain well or progress to sudden cardiac death, non-sudden cardiac death, non-cardiac death, unknown death.

Patients incur costs and effects by progressing through the model in monthly increments over a lifetime (420 months); a lifetime perspective allows the model to account for all costs incurred by patients that survive without a sudden cardiac arrest event. Patients in both treatment arms incur monthly inpatient and outpatient costs. In the ICD therapy arm, patients also incur the cost of the device and ICD implant procedure. ICD patients who remain alive long

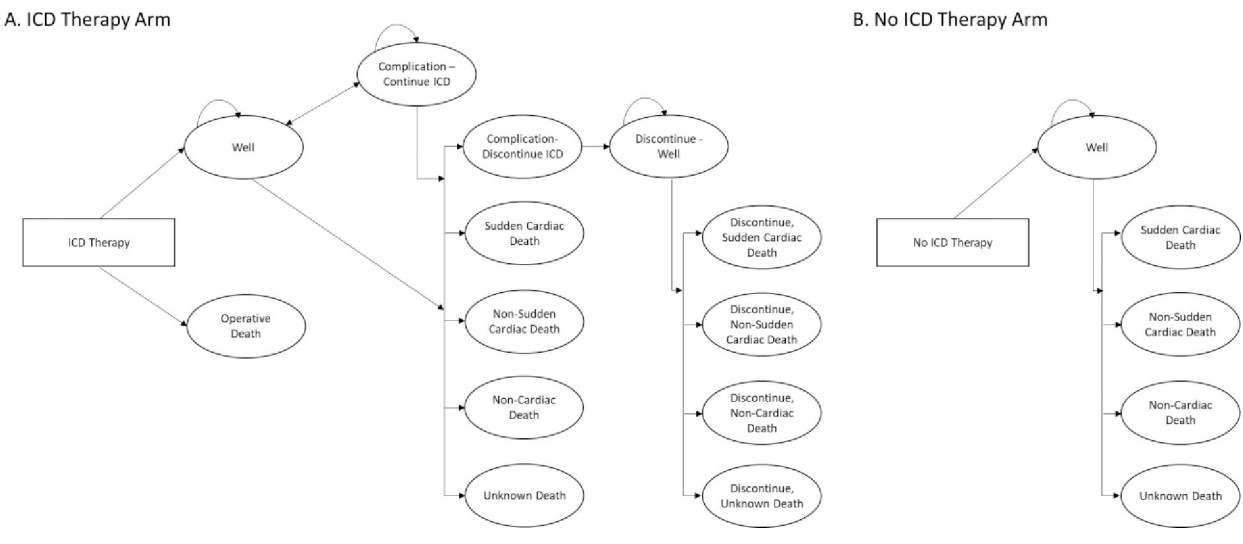

**Fig 1. Model schematic.**

enough to require a device replacement incur additional device and procedure costs at the time of replacement. ICD patients may receive an inappropriate shock or other ICD-related complication that incurs a cost and affects treatment adherence. After experiencing an inappropriate shock or other ICD-related complication, patients remain in the ICD therapy arm with ICD treatment or progress to discontinued use of ICD therapy. We assumed ICD patients who discontinue use of ICD therapy have the same mortality risk as patients in the no ICD arm.

## Clinical data

Mortality inputs to the model for primary prevention patients were based on a previously published meta-analysis and for 1.5 primary prevention patients were based on Improve SCA clinical study results. Primary prevention patients had a mean age of 61.1 years and were 76.3% male, while 1.5 primary prevention patients had a mean age of 61.1 years and were 79.5% male; other characteristics for each population are included in the original study publications [7, 11]. Other clinical inputs to the model for both ICD cohorts were based on the United States (US) National ICD registry, literature, and administrative claims-based analyses. The probability of implant-related operative death (0.0002) was based on the US National ICD Registry and applied only to the ICD treatment arms [12]. Inappropriate shock probability was derived from a weighted average based on the MADIT RIT, ADVANCE III, PROVIDE, and PainFree SST clinical trials that demonstrated a reduction in inappropriate shock rates due to device programming [13–16]. Probabilities of lead failure or dislodgement after initial implant were based on studies of annual incidence of lead failure and ICD lead dislodgement at one year after implant, 0.45% and 1.8% respectively [18, 19].  Probability of lead dislodgement or replacement after ICD replacement was based on data from the REPLACE registry that reported a 1% combined dislodgement and replacement rate [20]. We assumed half of the combined rate reported in the REPLACE registry could be attributed to lead failure (0.5%) and half could be attributed to lead dislodgement (0.5%). We estimated the one-year probability of lead infection after initial implant (1.22%) and after device replacement (2.16%) with a retrospective data analysis based on administrative claims from a large US insurer [21]. The lifetime risk of lead infection after the first year of an initial or replacement implant was double the value of the one-year claims-based probability [22].

## Economic data

Device related costs and long-term health care utilization costs associated with heart disease were modeled over a lifetime. Costs for individual events were assumed to be the same regardless of the indication (primary prevention or 1.5 primary prevention) for the ICD. The procedural costs of an initial ICD implant, subsequent revision or replacement, and ICD-related complications (infection and dislodgement) which including cost of devices, admission fee, drug fee, examination etc. were derived from the Taiwan Diagnosis-Related Group (Tw-DRG), edition 3.4 of the National Health Insurance Administration (NHIA), Ministry of Health and Welfare [23]. The procedural cost of inappropriate shocks and monthly long-term inpatient and outpatient costs were derived for the base case by evaluating the practice and calculated the cost base on "Fee Schedule for Medical Services of National Health Insurance" of the National Health Insurance Administration (NHIA), Ministry of Health and Welfare [24, 25]. Monthly inpatient and outpatient costs related to heart failure were estimated from the NHIA inpatient expenditures by admissions (DD) database and NHIA outpatient expenditures (OD) database, respectively.

## Health-related quality of life

Quality of life was based on an analysis of EQ-5D data collected in the PainFree SST clinical trial [28]. Taiwan-specific utilities were derived by mapping each patient's EQ-5D state using country specific societal preferences (S2 Table) [29]. We assumed the baseline utility for ICD patients and no ICD patients was the same [30]. Patients who experienced an ICD-related complication received a short-term utility decrement of 0.096 that is equivalent to 3.5 days [31].

## Construction of the ICER (w/WTP) and sensitivity analysis

Total lifetime costs and quality-adjusted life years (QALYs) between ICD therapy and no ICD therapy were simulated to calculate the incremental cost effectiveness ratio (ICER). Both undiscounted and discounted results were calculated to best represent the time value of costs and outcomes. We conducted one-way sensitivity and probabilistic sensitivity analyses to assess the impact of model inputs and parameter uncertainty. Beta distributions were used for probabilities and utilities in the probabilistic sensitivity analysis. We used a willingness-to-pay (WTP) threshold value of NT$2.1 million for this analysis [32].

# Results

## Base case scenario for primary prevention

Table 2 shows the results of the base-case scenario for primary prevention.

ICD therapy for primary prevention resulted in a benefit of 8.88 (discounted) and 9.82 (undiscounted) life-years saved, while no ICD therapy resulted in a benefit of 6.80 and 7.36 life-years saved, respectively. Measured in QALYs, the discounted benefit from ICD therapy is 6.48 and 4.98 from no ICD therapy, resulting in an incremental effectiveness of 1.51 QALYs. Discounted costs from ICD therapy and no ICD therapy account for NT$1,664,259 and NT$597,087, respectively. The ICER for ICD therapy is NT$708,711 per QALY; ICD therapy for primary prevention is cost-effective at NT$2.1 million.

## Base case scenario for 1.5 primary prevention

Table 2 (above) also shows the results of the base-case scenario for 1.5 primary prevention. ICD therapy for 1.5 primary prevention resulted in a benefit of 12.45 (discounted) and 14.17 (undiscounted) life-years saved, while no ICD therapy resulted in a benefit of 8.88 and 9.82 life-years saved, respectively. Measured in QALYs, the discounted benefit from ICD therapy is 10.78 and 7.71 from no ICD therapy, resulting in an incremental effectiveness of 3.08 QALYs. Discounted costs from ICD therapy and no ICD therapy account for NT$2,175,478 and NT$818,782, respectively. The ICER for ICD therapy is NT$441,153 per QALY; ICD therapy for 1.5 prevention is below one third of the WTP and is highly cost-effective at NT$2.1 million.

## Sensitivity analyses

Results of the sensitivity analyses are presented for the 1.5 primary prevention indication only (results for the primary prevention indication are in the S1 Table). Results of the one-way sensitivity analyses show that costs per QALY are more responsive to the age at implant, conventional mortality, replacement period and quality of life (Fig 2, Panel A). The incremental costs per QALY remained below the WTP threshold for all values of the one-way sensitivity analysis.

Fig 2, Panel B shows the simulated costs per QALY of the probabilistic sensitivity analysis. Results for 1.5 primary prevention show a mean cost per QALY of NT$434,053 (median cost

**Table 2. Base case scenario results (primary prevention and 1.5 primary prevention).**

| PP Base Case Scenario Results | | ICD therapy | No ICD Therapy |
|---|---|---|---|
| Undiscounted | Aggregated costs | NT$1,785,966 | NT$646,396 |
| | Differential cost | NT$1,139,570 | |
| | Effectiveness (life-years saved) | 9.82 | 7.36 |
| | Effectiveness (QALY saved) | 7.16 | 5.39 |
| | Differential effectiveness (QALY) | 1.77 | |
| | ICER (costs per QALY saved) | **NT$642,272** | |
| Discounted | Aggregated costs | NT$1,664,259 | NT$597,087 |
| | Differential cost | NT$1,067,172 | |
| | Effectiveness (life-years saved) | 8.88 | 6.80 |
| | Effectiveness (QALY saved) | 6.48 | 4.98 |
| | Differential effectiveness (QALY) | 1.51 | |
| | ICER (Costs per QALY saved) | **NT$708,711** | |
| 1.5PP Base Case Scenario Results | | ICD therapy | No ICD Therapy |
| Undiscounted | Aggregated costs | NT$2,410,603 | NT$905,881 |
| | Differential cost | NT$1,504,722 | |
| | Effectiveness (life-years saved) | 14.17 | 9.82 |
| | Effectiveness (QALY saved) | 12.27 | 8.53 |
| | Differential effectiveness (QALY) | 3.75 | |
| | ICER (costs per QALY saved) | **NT$401,722** | |
| Discounted | Aggregated costs | NT$2,175,478 | NT$818,782 |
| | Differential cost | NT$1,356,695 | |
| | Effectiveness (life-years saved) | 12.45 | 8.88 |
| | Effectiveness (QALY saved) | 10.78 | 7.71 |
| | Differential effectiveness (QALY) | 3.08 | |
| | ICER (Costs per QALY saved) | **NT$441,153** | |

**Abbreviations:** PP, Primary Prevention; ICD, Implantable Cardioverter-Defibrillator; QALY, quality-adjusted life year; ICER incremental cost-effectiveness ratio.

per QALY of NT$435,301, 95-percent Credible Interval [NT$291,938 –NT$1,115,321] per QALY) after 1,000 iterations. For 1.5 primary prevention, 99.5 percent of simulations result in costs per QALY below the WTP threshold, and 90 percent of simulations result in costs per

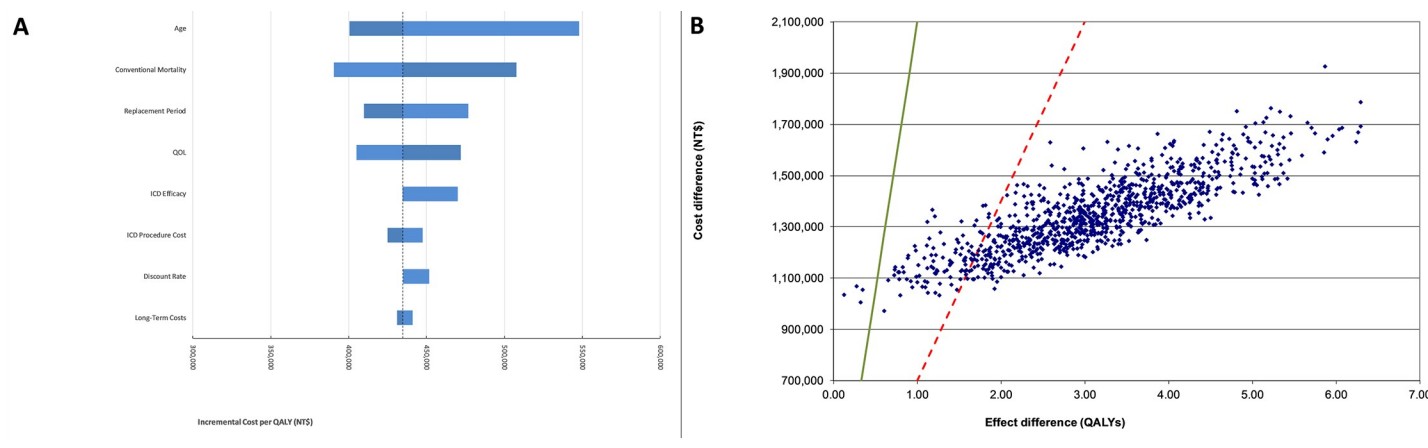

**Fig 2.**

QALY below one third of the WTP threshold, indicating ICD therapy is highly cost effective for this population.

## Discussion

Our results indicate that ICD therapy is cost effective for the whole primary prevention patient population and highly cost effective for the subset of 1.5 primary prevention patients in the Taiwan healthcare system. The primary prevention and 1.5 primary prevention are at ICERs of NT$708,711 per QALY and NT$441,153 per QALY respectively; while 1.5 primary prevention was more cost effective, both are less than the WTP value of NT$2.1 million. This finding is robust, with sensitivity analyses indicating that the cost effectiveness is preserved in nearly all reasonable variations of model inputs. To our knowledge, this is the first evaluation of the cost-effectiveness of ICD therapy compared to no ICD therapy among the whole primary prevention patient population and the subset of 1.5 primary prevention patients from the perspective of the public healthcare system in Taiwan.

Prior estimates of the cost effectiveness of ICD therapy have been performed in the primary prevention population. Mark et al [8] performed an analysis of the randomized SCD-HeFT trial and found ICD therapy to be economically attractive at $41,530/QALY (at a WTP of $100,000) in the US healthcare system. An analysis in the healthcare system of a European country using a meta-analysis of six randomized primary prevention trials and the same model used in this study showed similar results [7]. The cost-effectiveness of ICD therapy has also been confirmed in a real world setting outside of clinical trials [33]. Our study shows that ICD therapy for primary prevention patients cost less than 1 GDP per capita per QALY and appear even more cost-effective in the Taiwan healthcare system compared to previous reports in other countries (S1 Table).

Despite convincing evidence from multiple randomized clinical trials [1–3], and strong recommendations in international society guidelines [5, 6], ICD therapy remains underutilized in Asian countries [34, 35]. In particular, a report by Chia et al in 2017 [9] found that among guideline recommended ICD-eligible patients in Taiwan, only 7.7% had received ICD therapy, and by comparison, in Japan 52.5% of ICD-eligible patients had received ICD therapy [14]. National health expenditure (NHE) in Taiwan was 6.1% of GDP in 2017 [36] approximately one-third of the US (17.1%) and 69% of the average for OECD (Organization for Economic Cooperation and Development) countries (8.8%). Health spending per capita in 2017 in Taiwan was PPPUS$3,047 [36] less than one-third (30%) of the US total (PPPUS$10,209) and 76% of the average for OECD countries (PPPUS$3,992). To the extent that economic factors play a role, this study provides information for decision makers to direct scarce resources first toward those who can benefit the most. While it remains cost effective to treat the entire primary prevention population with ICD therapy, from an economic standpoint a priority could be placed on treating patients with a 1.5 primary prevention indication.

It is important to acknowledge the limitations of this analysis. The Improve SCA trial was not randomized, however the mortality benefit from the trial remained significant after adjusting for baseline characteristics likely to have an impact on mortality, and the effectiveness of ICD therapy for the primary prevention population has been shown to be replicated in both randomized and non-randomized observational trials. Costs and benefits were modeled beyond the timeline of direct observation in the Improve SCA trial, however this is a standard approach in economic modeling and necessary for the proper perspective for decision makers. For the full primary prevention analysis ICD effectiveness data were taken from a global meta-analysis that did not include Taiwan, however such data from Taiwan were not available. For the 1.5 primary prevention analysis patients in the Improve SCA trial were not all from

Taiwan, yet the majority were from Asia and ICD therapy application is well developed and largely standardized around the world. Conclusions from this report may not be generalizable beyond the Taiwan healthcare system.

## Conclusion

ICD therapy is cost effective for primary prevention patients in the Taiwan healthcare system, and highly cost effective for 1.5 prevention patients. These data provide guidance as to an efficient way to address underutilization of ICD therapy in indicated patients in Taiwan.

## Supporting information

**S1 Fig.**
(TIF)

**S2 Fig.**
(TIF)

**S1 Table. Characteristics and result of economic evaluations of ICD for primary prevention.**
(DOCX)

**S2 Table. Taiwan utility analysis results.**
(DOCX)

**S1 File. We have included a trace of the base case calculations for each patient population as a supplementary files, which give transparent and granular information on the calculations performed to produce the results.** This, together with the information in the manuscript (the specific model inputs and sources, and the structure of the model) is meant to allow for sufficient transparency to understand how the results were produced. S1 File, PP patients. S2 File, 1.5PP patients.
(XLS)

**S2 File. We have included a trace of the base case calculations for each patient population as a supplementary files, which give transparent and granular information on the calculations performed to produce the results.** This, together with the information in the manuscript (the specific model inputs and sources, and the structure of the model) is meant to allow for sufficient transparency to understand how the results were produced. S1 File, PP patients. S2 File, 1.5PP patients.
(XLS)

## Acknowledgments

We are grateful to Brian Van Dorn, MS for providing the utility estimates based on the IMPROVE SCA quality of life data and Janet E O'Brien, MS for medical writing assistance (from Medtronic).

## Author Contributions

**Conceptualization:** Reece Holbrook, Dave Phay, Yu-Cheng Hsieh, Kuo-Hung Lin, Yen-Bin Liu.

**Data curation:** Dave Phay.

**Formal analysis:** Reece Holbrook.

**Methodology:** Reece Holbrook.

**Supervision:** Yu-Cheng Hsieh, Kuo-Hung Lin, Yen-Bin Liu.

**Validation:** Lucas Higuera, Kael Wherry.

**Writing – original draft:** Reece Holbrook.

**Writing – review & editing:** Lucas Higuera, Kael Wherry, Dave Phay, Yu-Cheng Hsieh, Kuo-Hung Lin, Yen-Bin Liu.

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
