## [Decision Letter · Decision Letter 0]

19 Aug 2020

PONE-D-20-18597

Implantable cardioverter defibrillator therapy is cost effective for primary prevention patients in Taiwan: an analysis from the Improve SCA trial

PLOS ONE

Dear Dr. Yen Bin Liu

Thank you for submitting your manuscript to PLOS ONE. After careful consideration, we feel that it has merit but does not fully meet PLOS ONE’s publication criteria as it currently stands. Therefore, we invite you to submit a revised version of the manuscript that addresses the points raised during the review process.

We look forward to receiving your revised manuscript.

Kind regards,

Giuseppe Coppola

Academic Editor

PLOS ONE

Additional Editor Comments:

Kind authors, your paper underwent 3 different reviewers. Please read carefully comments and criticisms by statistic’s reviewer

Journal Requirements:

'The overall work was funded by Medtronic, plc. Mr. Holbrook, Mr. Higuera, Ms. Wherry, and Mr. Phay are Medtronic, plc. employees and stockholders. Drs. Hsieh, Lin, and I are consultants for Medtronic, Inc. and have not received compensation for the participation in this work.'

We note that one or more of the authors have an affiliation to the commercial funders of this research study: Medtronic.

2.1. Please provide an amended Funding Statement declaring this commercial affiliation, as well as a statement regarding the Role of Funders in your study. If the funding organization did not play a role in the study design, data collection and analysis, decision to publish, or preparation of the manuscript and only provided financial support in the form of authors' salaries and/or research materials, please review your statements relating to the author contributions, and ensure you have specifically and accurately indicated the role(s) that these authors had in your study. You can update author roles in the Author Contributions section of the online submission form.

2.2. Please also provide an updated Competing Interests Statement declaring this commercial affiliation along with any other relevant declarations relating to employment, consultancy, patents, products in development, or marketed products, etc.  

4. Please include your tables as part of your main manuscript and remove the individual files. Please note that supplementary tables (should remain/ be uploaded) as separate "supporting information" files.

Reviewers' comments:

Reviewer's Responses to Questions

**Comments to the Author**

1. Is the manuscript technically sound, and do the data support the conclusions?

Reviewer #1: Yes

Reviewer #2: Yes

Reviewer #3: No

2. Has the statistical analysis been performed appropriately and rigorously? 

Reviewer #1: Yes

Reviewer #2: Yes

Reviewer #3: No

3. Have the authors made all data underlying the findings in their manuscript fully available?

Reviewer #1: Yes

Reviewer #2: Yes

Reviewer #3: Yes

4. Is the manuscript presented in an intelligible fashion and written in standard English?

Reviewer #1: Yes

Reviewer #2: Yes

Reviewer #3: Yes

5. Review Comments to the Author

Reviewer #1: The paper shows a very interesting work that confirm the cost-effectiveness of ICD therapy. It's particularly interesting the sub-categorization of primary prevention patients. This sub-category includes patients at higher risk and therefore more likely to benefit from ICD implantation. In conclusion, a well-designed and interesting work although limited to the health reality of Taiwan

Reviewer #2: Excellent work, well written and structured, which further confirms the importance of ICD implantation in primary prevention of sudden cardiac death, also proving the cost-effectivenes of this practice in a particular subcategory of Taiwan population.

Reviewer #3: PONE-D-20-18597: statistical review

SUMMARY. This paper introduces a Bayesian network (called a Markov decision model by the authors) to evaluate the cost-effectiveness of Implantable cardiac defibrillators (ICD) compared to no ICD therapy for a

Taiwanese population at risk for sudden cardiac arrest (SCA). The network is initialized using input parameters obtained by multiple sources (Table 1) and exploited to estimate costs and life years saved (Table 2). Although the choice of a Bayesian network is potentially a correct method to evaluate cost-effectiveness, the paper lacks detailed methods (see major issues 1-5). Without these details, it is unclear whether the paper represents a technically sound and reproducible piece of scientific research.

MAJOR ISSUES

1. Study population. The study population of the primary prevention patients is not defined. A table should be provided with the main characteristics of the subjects under analysis.

2. Input parameters. Parameter values are obtained from multiple sources with different degrees of uncertainty (see the standard errors of Table 1). How were these standard errors integrated in the analysis?

3. Estimation. Nothing is said about the estimation procedure exploited to obtain the output of Table 2. How were these estimates obtained? In addition, standard errors should be provided along with the estimates. The output of a Bayesian network is typically the probability distribution of the output variables, given the inputs. What do the outputs of Table exactly represent? The expectation of the probability distribution? Or its mode?

4. The excel sheet that has been exploited to obtain the results should be provided as a supplementary information file, to guarantee reproducibility.

5. Details about the sensitivity analysis should be provided. How many times was the network simulated? Which values of the variables of Figure 2, panel A, were exploited?

SPECIFIC COMMENTS

1. Table 1 includes a strange statement "Error! Reference source not found". Please check. In addition: what is the "beta distribution" mentioned in the table?

2. Figure 1. Transition probabilities should be attached to each arc of the graph.

6. PLOS authors have the option to publish the peer review history of their article (what does this mean?). If published, this will include your full peer review and any attached files.

Reviewer #1: **Yes: **Gianfranco Ciaramitaro, M.D. Ph.D

Reviewer #2: No

Reviewer #3: No

---

## [Author Response · Author response to Decision Letter 0]

31 Aug 2020

Dear Reviewers, 

Thank you for your thoughtful review and questions. Please refer to the "Reviewer Response Letter" for the following responses to your questions. 

1. Study population. The study population of the primary prevention patients is not defined. A table should be provided with the main characteristics of the subjects under analysis.

The study populations (both primary prevention and 1.5 primary prevention) are not new cohorts, they have been previously published. We have added general characteristics to this manuscript and given a clear reference to the publications where more detailed characteristics can be found. The following sentence was added to the methods section of the manuscript in the subheading “Clinical Data”: Primary prevention patients had a mean age of 61.1 years and were 76.3% male, while 1.5 primary prevention patients had a mean age of 61.1 years and were 79.5% male; other characteristics for each population are included in the original study publications.

2. Input parameters. Parameter values are obtained from multiple sources with different degrees of uncertainty (see the standard errors of Table 1). How were these standard errors integrated in the analysis?

The analysis was done in two parts, following standard cost-effectiveness analysis methodology (Ramsey SD, Willke RJ, Glick HA, et al. Cost-effectiveness analysis alongside clinical trials II: an ISPOR Good Research Practices Task Force Report. Value Health. 2015;18(2):161-172). 

The first part is a deterministic base case analysis, in which the base case inputs from Table 1 are entered into the model as point estimates and the ICER is calculated. The result of the base case analysis, which does not use the standard error values, is documented in Table 2. This ICER is deterministic and does not account for any variability and results in a single numeric point estimate for the ICER. The following sentence in the methods section was modified to make this clearer:

Total lifetime costs and quality-adjusted life years (QALYs) between ICD therapy and no ICD therapy were simulated using base case model inputs to calculate the deterministic incremental cost effectiveness ratio (ICER).

After completion of the deterministic base case analysis, we performed 2 different types of sensitivity analyses. The first is the one-way sensitivity analysis, which is also deterministic, but varies a single input to its high and low values while holding all other inputs constant resulting in a set of numeric point estimate ICER values which are represented in Figure 2, panel A. We updated the following sentence in the methods section to more clearly indicate that the one-way sensitivity analysis is deterministic (the probabilistic sensitivity analysis is already aptly named):

We conducted deterministic one-way sensitivity and probabilistic sensitivity analyses to assess the impact of model inputs and parameter uncertainty.

The second sensitivity analysis was a probabilistic sensitivity analysis, where input parameters were all varied simultaneously by selecting random values from a probability distribution based on the mean and standard error values of the parameters. This analysis produced the findings in the final paragraph of the results section: 

Results for 1.5 primary prevention show a mean cost per QALY of NT$434,053 (median cost per QALY of NT$435,301, 95-percent Credible Interval [NT$291,938 – NT$1,115,321] per QALY) after 1,000 iterations. For 1.5 primary prevention, 99.5 percent of simulations result in costs per QALY below the WTP threshold, and 90 percent of simulations result in costs per QALY below one third of the WTP threshold, indicating ICD therapy is highly cost effective for this population.

3. Estimation. Nothing is said about the estimation procedure exploited to obtain the output of Table 2. How were these estimates obtained? In addition, standard errors should be provided along with the estimates. The output of a Bayesian network is typically the probability distribution of the output variables, given the inputs. What do the outputs of Table exactly represent? The expectation of the probability distribution? Or its mode?

As stated in the response to question #2 above, the results in Table 2 represent a single point estimate and are considered the base case results per standard cost effectiveness analysis methodology. Uncertainty and variability of the model are addressed with the sensitivity analyses which are also described in the response to question #2 above.

4. The excel sheet that has been exploited to obtain the results should be provided as a supplementary information file, to guarantee reproducibility.

We have included a trace of the base case calculations for each population as supplementary files, which gives transparent and granular information on the calculations performed to produce the results. This, together with the information in the manuscript (the specific model inputs and sources, and the structure of the model) is meant to allow for sufficient transparency to understand how the results were produced.

5. Details about the sensitivity analysis should be provided. How many times was the network simulated? Which values of the variables of Figure 2, panel A, were exploited?

Some of these details, including how many times the network was simulated, are included in the manuscript: 

Results for 1.5 primary prevention show a mean cost per QALY of NT$434,053 (median cost per QALY of NT$435,301, 95-percent Credible Interval [NT$291,938 – NT$1,115,321] per QALY) after 1,000 iterations. For 1.5 primary prevention, 99.5 percent of simulations result in costs per QALY below the WTP threshold, and 90 percent of simulations result in costs per QALY below one third of the WTP threshold, indicating ICD therapy is highly cost effective for this population.

The variables that could change in the probabilistic sensitivity analysis were the variables that included standard error values in Table 1. The following sentence was added to the last paragraph in the methods section to make that clearer for the reader:

For the probabilistic sensitivity analysis, the inputs that were varied were the ones with standard errors reported in Table 1, and for each run of the model those inputs were randomly varied according to a beta probability distribution based on the base case and standard error values of the corresponding input.

SPECIFIC COMMENTS

1. Table 1 includes a strange statement "Error! Reference source not found". Please check. In addition: what is the "beta distribution" mentioned in the table?

Thanks for pointing out the missing reference error message, it is now corrected. Regarding the beta distribution, see the answer to comment #5 above.

2. Figure 1. Transition probabilities should be attached to each arc of the graph.

It is not common for publications on cost effectiveness of medical therapies to include transition probabilities in the figure on the model structure (see references included below). There are many probabilities, and they are quite precise, so inclusion on the graph directly would cloud understanding of the general state transitions and pathways inherent to the model structure, which is the purpose of the figure. There is full transparency in the manuscript, as the transition probabilities are all included in Table 1 in the description of model inputs.

Representative publications on the cost effectiveness of medical therapies that include figures on model structure while reporting transition probabilities separately:

• Sanders GD et al. Cost-Effectiveness of Implantable Cardioverter–Defibrillators. N Engl J Med. 2005 Oct 6;353(14):1471-80.

• Gada H et al. Markov Model for Selection of Aortic Valve Replacement Versus Transcatheter Aortic Valve Implantation (Without Replacement) in High-Risk Patients. Am J Cardiol. 2012 May 1;109(9):1326-33.

• Canestaro WJ et al. Cost-Effectiveness of Oral Anticoagulants for Treatment of Atrial Fibrillation. Circ Cardiovasc Qual Outcomes. 2013 Nov;6(6):724-31

Dear Reviewers, the cover letter, ethics statement, manuscript and reviewer response letter has been updated to address you questions included in your 29 August 2020 email. Please review these revisions. Thank you, Dr Liu

---

## [Decision Letter · Decision Letter 1]

20 Oct 2020

Implantable cardioverter defibrillator therapy is cost effective for primary prevention patients in Taiwan: an analysis from the Improve SCA trial

PONE-D-20-18597R1

Dear Dr. Liu,

We’re pleased to inform you that your manuscript has been judged scientifically suitable for publication and will be formally accepted for publication once it meets all outstanding technical requirements.

Kind regards,

Andrea Ballotta

Academic Editor

PLOS ONE

Additional Editor Comments (optional):

Congratulations nulla obsta for the publication of the above mentioned manuscript.

Reviewers' comments:

Reviewer's Responses to Questions

**Comments to the Author**

1. If the authors have adequately addressed your comments raised in a previous round of review and you feel that this manuscript is now acceptable for publication, you may indicate that here to bypass the “Comments to the Author” section, enter your conflict of interest statement in the “Confidential to Editor” section, and submit your "Accept" recommendation.

Reviewer #1: (No Response)

Reviewer #3: All comments have been addressed

2. Is the manuscript technically sound, and do the data support the conclusions?

Reviewer #1: (No Response)

Reviewer #3: (No Response)

3. Has the statistical analysis been performed appropriately and rigorously? 

Reviewer #1: (No Response)

Reviewer #3: (No Response)

4. Have the authors made all data underlying the findings in their manuscript fully available?

Reviewer #1: (No Response)

Reviewer #3: (No Response)

5. Is the manuscript presented in an intelligible fashion and written in standard English?

Reviewer #1: (No Response)

Reviewer #3: (No Response)

6. Review Comments to the Author

Reviewer #1: (No Response)

Reviewer #3: (No Response)

7. PLOS authors have the option to publish the peer review history of their article (what does this mean?). If published, this will include your full peer review and any attached files.

Reviewer #1: No

Reviewer #3: No

---

## [Editor Report · Acceptance letter]

30 Oct 2020

PONE-D-20-18597R1 

Implantable cardioverter defibrillator therapy is cost effective for primary prevention patients in Taiwan: an analysis from the Improve SCA trial 

Dear Dr. Liu:

I'm pleased to inform you that your manuscript has been deemed suitable for publication in PLOS ONE. Congratulations! Your manuscript is now with our production department. 

Kind regards, 

on behalf of

Dr. Andrea Ballotta 

Academic Editor

PLOS ONE